# Echocardiography and Electrocardiography in Detecting Atrial Cardiomyopathy: A Promising Path to Predicting Cardioembolic Strokes and Atrial Fibrillation

**DOI:** 10.3390/jcm12237315

**Published:** 2023-11-26

**Authors:** Delicia Gentille-Lorente, Alba Hernández-Pinilla, Eva Satue-Gracia, Eulalia Muria-Subirats, Maria Jose Forcadell-Peris, Jorge Gentille-Lorente, Juan Ballesta-Ors, Francisco Manuel Martín-Lujan, Josep Lluis Clua-Espuny

**Affiliations:** 1Servicio de Cardiología, Hospital Verge de la Cinta, 43500 Tortosa, Spain; 2Institut Investigacio Sanitaria Pere Virgili (IISPV), 43204 Reus, Spain; 3Primary Care Service Camp de Tarragona, Institut Català de la Salut, 43005 Tarragona, Spain; ahernandez.tgn.ics@gencat.cat; 4Research Support Unit, Fundació Institut Universitari per a la Recerca a l’Atenció Primària de Salut Jordi Gol i Gurina (IDIAPJGol), 43202 Reus, Spain; esatue.tgn.ics@gencat.cat (E.S.-G.); fmartin.tgn.ics@gencat.cat (F.M.M.-L.); 5Primary Health-Care Centre, Institut Català de la Salut, Primary Care Service (SAP) Terres de l’Ebre, EAP Amposta, 43870 Amposta, Spain; eumuria@gmail.com (E.M.-S.); mjforcadell.tgn.ics@gencat.cat (M.J.F.-P.); 6Primary Health-Care Centre, Institut Català de la Salut, Primary Care Service (SAP) Terres de l’Ebre, UUDD Tortosa-Terres de l’Ebre, 43500 Tortosa, Spain; jgentille.ebre.ics@gencat.cat; 7Emergencies and Special Care Department, Catholic University of Murcia, Medical Management 061, 30107 Guadalupe, Spain; jballest@ucam.edu; 8Primary Care Service Campus Tarragona, University Rovira Virgili, 43002 Tarragona, Spain; 9Primary Health-Care Centre, Institut Català de la Salut, Primary Care Service (SAP) Terres de l’Ebre, 43500 Tortosa, Spain; 10Research Support Unit, Fundació Institut Universitari per a la Recerca a l’Atenció Primària de Salut Jordi Gol i Gurina (IDIAPJGol), 43500 Tortosa, Spain

**Keywords:** atrial cardiomyopathy, atrial fibrillation, interatrial block, advanced interatrial block, Bayés’s Syndrome, atrial strain, reservoir strain, electrocardiogram, transthoracic echocardiogram, cardioembolic stroke, embolic stroke of unknown source

## Abstract

(1) Background: Atrial cardiomyopathy constitutes an intrinsically prothrombotic atrial substrate that may promote atrial fibrillation and thromboembolic events, especially stroke, independently of the arrhythmia. Atrial reservoir strain is the echocardiography marker with the most robust evidence supporting its prognostic utility. The main aim of this study is to identify atrial cardiomyopathy by investigating the association between left atrial dysfunction in echocardiography and P-wave abnormalities in the surface electrocardiogram. (2) Methods: This is a community-based, multicenter, prospective cohort study. A randomized sample of 100 patients at a high risk of developing atrial fibrillation were evaluated using diverse echocardiography imaging techniques, and a standard electrocardiogram. (3) Results: Significant left atrial dysfunction, expressed by a left atrial reservoir strain < 26%, showed a relationship with the dilation of the left atrium (*p* < 0.001), the left atrial ejection fraction < 50% (*p* < 0.001), the presence of advanced interatrial block (*p* = 0.032), P-wave voltage in lead I < 0.1 mV (*p* = 0.008), and MVP ECG score (*p* = 0.036). (4) Conclusions: A significant relationship was observed between left atrial dysfunction and the presence of left atrial enlargement and other electrocardiography markers; all of them are non-invasive biomarkers of atrial cardiomyopathy.

## 1. Introduction

Atrial fibrillation (AF) is the most prevalent sustained cardiac arrhythmia in adults worldwide. Its prevalence is expected to increase two to three times due to the extended lifespan of the general population and the growing efforts to identify undiagnosed cases of AF [1]. AF is associated with 25–36% of all strokes [2,3,4]. On the other hand, most cryptogenic strokes are embolic strokes of undetermined source (ESUS), which already represent at least 25% of all ischemic strokes [5]; moreover, AF and cardioembolic strokes often coexist and can share common pathophysiologic mechanisms.

The term “atrial cardiomyopathy (AC)” has been introduced recently, defined as any complex of structural, architectural, contractile, or electrophysiological changes affecting the atria with the potential to produce clinically relevant manifestation [6]. Strong evidence suggests that AC constitutes an intrinsically prothrombotic atrial substrate that may precede and promote AF and thromboembolic events independent of the arrhythmia [7,8]. Therefore, there is a compelling need to identify patients with AC in a preclinical stage, meaning before the detection of arrhythmias or the development of embolic ischemic strokes. AC can be detected through non-invasive methods such as electrocardiography (ECG), transthoracic echocardiogram (TTE), and cardiac magnetic resonance imaging (MRI), or invasively through high-density electro-anatomical studies [9]. According to the definition provided above, any change in the dimensions and/or function of the left atrium (LA) can be classified as AC. These disruptions can manifest in a wide range of pathological conditions as well as in physiological states. Therefore, it is essential to develop non-invasive and effective methods for its detection and the potential modification of risk factors or treatment. 

Concerning the ECG, numerous parameters are linked to both LA enlargement and dysfunction [10,11,12,13,14,15,16,17,18,19,20,21]. The most extensively studied P-wave indices encompass the interatrial blocks (IAB), the P-wave terminal force in V1 (PTFV1), and P-wave axis and voltages (Appendix A) [21,22,23,24,25].Highlight that the advanced interatrial block (A-IAB) is regarded as an indicator of an atrial fibrosis [13] and, consequently, a marker of the electromechanical LA dysfunction; by this, is a marker of AC and a important predictor of, mainly, supraventricular arrhythmias (Bayes’ Syndrome), cardioembolic stroke and dementia [6,13,17,26,27].

Among TTE parameters, an increased LA volume and reduced LA ejection fraction (LAEF) have been established as independent predictors of incident AF, stroke, and other adverse cardiovascular events [6]. Recently, new echocardiographic techniques of two-dimensional speckle tracking have been employed to assess LA function [28] by means myocardial strain imaging. The positive systolic peak value of the global LA strain curve, during the LA reservoir phase (reservoir strain value), is the parameter that has emerged as the biomarker with the most robust evidence supporting its diagnostic and prognostic utility [10,29,30] across diverse clinical conditions and populations; it is considered as an indicator of atrial fibrosis and structural remodeling [31,32,33].

Indeed, the EACVI/EHRA Expert Consensus Document for assessing patients with AF emphasizes the significance of utilizing LA strain studies as a valuable adjunctive tool in this context. Furthermore, the updated European AF guidelines [1] underscore the importance of incorporating an LA strain analysis for a more precise evaluation of LA function. Recent findings have indicated that three-dimensional TTE (3D-TTE) may offer superior accuracy and correlation with MRI compared to conventional 2D-TTE when measuring LA volumes and LAEF. Although TTE is the major non-invasive diagnostic tool for real-time imaging of cardiac structure and function, our hypothesis suggests that the standard ECG might be an efficient and cost-effective tool to detect early AC among at-risk populations, due to its widespread availability, and could potentially play a significant role in stroke prevention.

The Action Plan in Europe (2018–2030) prioritizes the availability of detection and treatment programs in primary care to improve the diagnosis and monitoring of populations at risk of AF [1]. The primary objective of this study was to explore the relationship between LA size and function, as assessed through 2D-TTE and 3D-TTE, and the presence of of P-wave abnormalities in standard ECG.

## 2. Materials and Methods

This is a community-based, multicenter, prospective cohort study of a population aged ≥ 65 years. It included a randomized sample of patients enrolled in the ongoing PREFATE study [34], which is a project that emerged as a continuation of the results of the AFRICAT project (NCT03188484) [35].

### 2.1. Target Population 

A randomized sample of 100 patients aged ≥ 65 years were recruited from the usual consultations in primary healthcare centers managed by the Catalan Health Institute, and located in Tarragona region (South Catalonia, Spain). All of them were at a high risk of developing AF according to the AF-risk calculator, based on a previously validated predictive model for AF in the general population [36]. The follow up will span 2 years, from 1 January 2023 to 31 December 2024. This study presents baseline results (Figure 1, flowchart) obtained through standard ECG and TTE. To detect a new diagnosis of AF with a difference equal to or greater than 0.1 units compared to the reference population, a sample of 100 individuals will be required, accepting an alpha risk of 0.05 and a beta risk of 0.20 in a bilateral test. It is assumed that the proportion in the reference group is 0.10. A follow-up loss rate of 15% was estimated.

### 2.2. Selection Criteria

Selection criteria are (1) an age of 65–85 years; (2) Q4 according to the AF-risk calculator based on a previous validated predictive model for AF in the general population [36]; (3) no prior AF or stroke; (4) a CHA2DS2-VASc score ≥ 2; (5) ability to manage data using a mobile device [37,38] either by the patient or caregiver; and (6) informed consent. 

### 2.3. Variables

The dependent variables are LA size and function evaluated by 2D-TTE and 3D-TTE; P-wave abnormalities in the standard ECG; and a new diagnosis of AF (any type). The independent variables consist of the following: variables for AF risk calculation, encompassing sociodemographic factors, clinical variables, comorbidities, clinical scores, and active treatment (see Table 1).

### 2.4. Intervention

#### 2.4.1. Basal Cardiac Rhythm Monitoring

Clinical follow up and cardiac rhythm monitoring were performed for 15 days using cardiac rhythm monitoring devices, Fibricheck^R^ [37,38]. A referring cardiologist will evaluate and validate the records as well as the diagnosis of AF and the presence and intensity of supraventricular arrhythmias. The diagnosis of AF was confirmed with ECG and/or Holter monitoring for at least 5 days. Baseline data were collected in an anonymized fashion, automatically when possible or manually, from medical professional/hospital databases (E-cap, EQA, SIRE), as well as through devices. These datasets were described previously [36].

#### 2.4.2. Electrocardiogram Study

A standard surface 12-lead ECG (with a filter setting of 150 Hz, recording speed of 25 mm/s, and calibration of 10 mm/mV) was conducted for all patients. An experienced cardiologist, who was blinded to the patients’ medical history and TTE data, analyzed all ECG recordings by enlarging them using a zoom tool. A P-wave analysis was manually performed using the six frontal leads, displayed simultaneously, to measure P-wave duration in milliseconds (ms). The duration was determined by measuring the interval between the onset of the P-wave (in the initial lead where it first appeared) and its conclusion (in the lead where it subsequently disappeared). P-wave morphology was also assessed. The presence of an interatrial block (IAB) was determined according to the criteria established in the consensus documents [22,39]. Additionally, P-wave voltage in lead I (in millivolts) (PVI) and the duration of the P-terminal force in V1 (in milliseconds) (PTFV1) were analyzed. Other included parameters were the presence of an atrioventricular block (based on PR interval), MVP score (Morphology-Voltage-P-wave duration) (Appendix A), the amplitude of the QRS complex, and the Sokolow-Lyon and Cornell indexes to assess left ventricular hypertrophy.

#### 2.4.3. Echocardiogram Study

A TTE was performed using the EPIQ 7 ultrasound system and an X5-1 transducer (Philips Medical Systems, Amsterdam, the Netherlands). TTE examinations were performed and analyzed by a lone, experienced echo-cardiographer who remained unaware of the patients’ medical history and ECG parameters (Table 2). The assessments included conventional 2D-TTE, followed by the 2D-speckle tracking technique and 3D-TTE images. All studies were digitally stored and analyzed offline using IntelliSpace Cardiovascular QLAB 15.0 software (IISCV-QLAB) from Philips Medical Systems, Amsterdam, the Netherlands.

Following the guidelines for cardiac chamber assessment [39], LA volumes were assessed using apical four- and two-chamber images at the end of left ventricular (LV) systole with care taken to avoid foreshortening of the LA. The endocardial borders of the LA were delineated, excluding the junctions of the pulmonary veins and the LA appendage. The indexed 2D volume was calculated using the biplane disk summation method and the patient body surface. LAEF was determined using the Simpson method from both apical four- and two-chamber views. Other conventional TTE parameters were recorded in accordance with the current recommendations of the imaging societies [40,41,42].

In line with the consensus document from EACVI/ASE/Industry [43], 2D-speckle tracking was conducted using a non-foreshortened apical four-chamber LA view, with a frame rate ranging from 50 to 70 frames per second. These images were digitally stored in cine-loop format. The region of interest used for the analysis was the endocardial LA border, which was initially traced automatically, with IISCV-QLAB, and then subjected to manual editing to ensure optimal results. This manual editing involved verifying the tracking accuracy by comparing the superimposed trace results with the underlying image loop. The endocardial LA borders were defined to exclude the pulmonary veins and LA appendage orifices. Strain measurements for the LA reservoir, conduit, and contractile phases were calculated with the timing of the initial onset of the QRS complex (LV end diastole) serving as the zero-baseline point for the LA strain curves.

Lastly, a multiple-beat real-time 3D-TTE full volume dataset was obtained from the apical four-chamber view, with the patient holding their breath. Entirely automated heart model software was applied in real-time, and this was followed by manual regional editing using a semi-automated approach to guarantee the best results. The 3D LA index volumes and 3D LAEF values were then recorded and stored.

### 2.5. Abnormal Range of LA Strain

Based on the evidence [29,30,44,45,46], a LA reservoir strain (LA-Sr) value < 26% offers robust prognostic information regarding the risk of AF and/or ischemic stroke in various populations. It was assumed that the group of patients with LA-Sr < 26% exhibited significant atrial dysfunction, and consequently, an increased risk of adverse cardiovascular events. Thus, this series of patients was divided into two groups based on LA-Sr: <26% and ≥26%.

### 2.6. Statistical Analysis

The data were presented using frequencies and percentages for categorical variables and means with standard deviations for continuous variables, categorized by LA-Sr (<26% vs. ≥26%). To assess differences between the two groups, the chi-square test was applied to categorical variables, the Mann–Whitney U test was applied to continuous variables, and bivariate correlational analyses were conducted to explore the relationship between ECG and TTE variables. Multinomial logistic regression was employed to examine the association between the outcome (LA-Sr < 26% as the dependent variable) and predictor variables, along with determining the significance of each predictor in this relationship. The results were reported as odds ratios (ORs) with corresponding 95% confidence intervals. The relationship between variables in the logistic regression model was evaluated using the area under the curve (AUC) of the receiver operating characteristic (ROC) curve. All analyses were performed using the statistical package R (R Foundation for Statistical Computing, Vienna, Austria; version R 3.4.3 for Windows).

## 3. Results

### 3.1. Baseline Features

Baseline features are displayed in Table 1. The average age was 74.9 ± 5.3, and 57% of the participants were women. In this series, 100 patients underwent ECG, 93 accepted and underwent TTE. Ultimately, 86 people were enrolled because speckle tracking and 3D-TTE techniques could not be performed due to suboptimal echocardiographic access. 

### 3.2. Echorcardiography Study 

The results presented in Table 2 demonstrate a relationship between LA dysfunction, expressed by an LA-Sr < 26%, and the presence of a dilated LA indexed biplane volumen (LA-ibV ≥ 34 mL/m^2^; *p* < 0.001) and hypocontractile LA in the 2D-TTE (2D-LAEF < 50%; *p* < 0.001) and in the 3D-TTE (3D-LAEF < 55%; *p* = 0.001) (Figure 2). LA dilation was presented in 48.4% of the patients (*p* = 0.004), with an average LA-ibV of 36.4 mL/m^2^ (*p* < 0.001). However, when breaking down the results based on the degree of dilation, it was observed that the dilation was mild in 60% of cases (compared to 66.7% in the group with LA-Sr ≥ 26%) and severe in 33.3% of cases (compared to 11.1% in the group with LA-Sr ≥ 26%). LA dysfunction did not exhibit a significant relationship neither left valvular disease nor LV dysfunction variables.

### 3.3. Standard ECG Variables 

The presence of advanced IAB (A-IAB) (*p* = 0.032), a PVI < 0.1 mV (*p* = 0.007), and a major MVP ECG score (“*original*”, *p* = 0.038; “*modified*”, *p* = 0.036) were found to be associated with LA dysfunction expressed by an LA-Sr < 26% (Table 3). A-IAB was identified in 26 patients (30.2%), with 18 (69.2%) cases classified as typical A-IAB and 8 (9.3%) as atypical A-IAB. The presence of A-IAB exhibited a notable association with LA dysfunction, expressed as LA-Sr < 26% (*p* = 0.032).

Patients with an LA-Sr < 26% were mainly those with an A-IAB plus PVI < 0.1 mV (Figure 3).

In the baseline evaluation, seven new cases of AF were detected using the Fibricheck^R^ app, all of which were in the LA Sr < 26 group (*p* < 0.001). As shown in Figure 4, it can be observed that all of them are in the high-risk group according to the MVP score and have the lowest LA-Sr.

The regression model results identified as independent prognostic factors, the following variables: LA-ibV ≥ 34 mL/m^2^ [OR: 5.2, CI95%: 1.30–21.2; *p* = 0.020], 2D-LAEF < 50% [OR: 30.6, CI 95%: 8.1–115.0; *p* < 0.001], and A-IAB [OR: 6.9, CI95%: 0.72–67.7; *p* = 0.094], being the variables with the most discriminative capacity: 2D-LAEF (AUC: 0.84, CI95%: 0.74–0.94; *p* < 0.001], 2D LA-ibV (AUC: 0.67, CI95%: 0.55–0.80; *p* = 0.008), and A-IAB (AUC: 0.60, CI95%: 0.47–0.73; *p* = 0.109) (Figure 5).

## 4. Discussion

The study results underscored a notable connection between ECG P-wave abnormalities and LA size and function parameters assessed through 2D-TTE and 3D-TTE. Integrating LA strain imaging with standard ECG holds promise in paving the way for an early atrial fibrillation diagnosis and clinical management, and for prevent cardioembolic stroke irrespective of heart rhythm. LA size and function can change and be altered independently of each other, particularly in cases where the LA is only slightly dilated; these LA abnormalities serve as predictive parameters for outcomes across various physiological and pathological scenarios; that is because both are considered non-invasive markers of the atria fibrosis and the electrical and structural remodeling, that is, markers of AC. 

### 4.1. Relationship between Reduced Strain-R < 26% and the Rest of TTE Parameters

When LA function is evaluated by means 2D-TTE speckle tracking techniques it is obtained the LA strain curves (as the result of LA myocardial deformation expresed as a percentage). LA systolic and diastolic function is subdivided into three consecutive phases of a cyclic process: (a) reservoir phase, that corresponds to the reservoir strain (LA-Sr), (b) conduit phase, that corresponds to the conduit strain (LA-Scd), and (c) contractile or pump phase, that corresponds to the contraction strain (LA-Sct) [28]. The positive systolic peak value of the global LA strain curve, with zero as the reference, corresponds to the peak of LA-Sr; this is the parameter with the most robust evidence supporting its diagnostic and prognostic utility [10,29,30] across diverse clinical conditions and populations, because is considered an indicator of atrial fibrosis and structural remodeling [31,32,33]. A consensus reached through a meta-analysis of a large number of healthy control subjects has established a normal reference range (Appendix A) [32]. LA-Sr values of ≤25.8% [47,48] have been documented as indicators for identifying AF after cryptogenic strokes, and an elevated risk of stroke recurrence and mortality [11]. In the general population, other authors found that LA-Sr < 23% and 2D-LAEF < 55% were independently associated with an increased risk of cardiac events and new-onset AF [44]. For individuals under 65 years old, a LA-Sr value < 31.2% was associated with the risk of AF and ischemic stroke [45]. Several authors describe that the LA strain depends mainly on the age [12,13] and, therefore, they describe establishing age-associated normal values that may assist in distinguishing “healthy” LA aging from pathological processes [32].

Based on these foundings of previous studies, the threshold for “normality” was defined as LA-Sr ≥ 26%, leading to the categorization of two patient groups. Among them, 36% exhibited an LA-Sr value of <26%, but this specific condition did not show associations with age, sex, or traditional cardiovascular risk factors, including diabetes, hypertension, or chronic kidney disease, nor did it correlate with a history of heart failure or ischemic cardiopathy (Table 1). Not even the CHA2DS2-VASc score revealed differences among patients with an LA-Sr < 26%, emphasizing the potential limitations in discerning thrombogenic risk based on the evidence in the literature regarding the predictive power of ECG and TTE data [10,11,12,13,14,15,16,17,18,29,30,45,46,47,48].

Similarly, a notable association was observed between a reduced LA-Sr < 26% and a reduced Sct (<8.4%; *p* < 0.001). Consistent with this, limited evidence indicates that values < 17.4% are considered abnormal [13], and values < 9.6% predict AF [22]. Considering the other TTE parameters, in accordance with the recommendations for Cardiac Chamber Quantification by Echocardiography [22], the upper normal limit for 2D-TTE LA-ibV was set at 34 mL/m^2^ for both genders. This adjustment accounts for gender-dependent differences in LA size, generally factored in when adjusting for body size (Appendix A). Regarding LA size, the results are similar to those described by other authors previously [40]: LA dilation is associated with the dysfunction and, although both disturbances are independents, in general, worse strain values tend to correspond to larger LA sizes (33.3% of cases in LA-Sr < 26% group compared to 11.1% in the group with LA-Sr ≥ 26%).

Another indicator of LA global function is the LAEF. Values under 50% have been associated with cardioembolic stroke and paroxysmal AF [49,50]. As indicated by the findings, LA-Sr < 26% demonstrated a significant association with a reduced LAEF in 2D-TTE (with an average of 42.4%; *p* < 0.001).

Respect to the newer 3D-TTE imaging, its know that 3D-TTE LA volumemes are typically larger than 2D-TTE, and LA size and function correlates well with MRI; howewer, at this time, 3D-TTE techniques have importants limitations because of the lack of a standardized methodology, the indeterminated normal parametres values, the limited availability to the 3D-TTE imaging technology, and the imposibility to analizy 3D images in case of suboptim echocardiografic patient acces. A recent systematic review and meta-analysis has provided that nomal LA maximum volume indexed is <25.18 mL/m^2^ and nomal LAEF is ≥55.94% [19]. In the results, as previously described, it was evident that LA-Sr < 26% was significantly associated with a lower LAEF in 3D-TTE (with an average of 44.7%; *p* < 0.001), and there was a clear trend towards an association with a larger 3D-TTE LA volume as well; however, was necessary a considerable editing image to obtain the most precise values.

In both the ROC curve and multiple regression analysis, LAEF emerges as the most significant independent factor linked to an LA-Sr < 26%, likely attributable to its connection with LA function.

Finally, it is worth noting that LA strain mechanics are affected by left ventricular (LV) diastolic dysfunction before the onset of functional and structural changes in the LV. However, in this series, there is no observed association with LV diastolic dysfunction (including grades II-IV), LV systolic function (defined as LV ejection fraction < 50% or even <40%), and the presence of significant left valvular disease.

### 4.2. Relationship between Reduced LA-Sr < 26% and ECG Parameters

Several P-wave abnormalities were points of interest in the present study. A-IAB had a stronger predictive value for supraventricular arrhythmiasin many clinical scenarios (mainly atrial flluter and AF, wich is denominated Bayés’ Syndrome), for cardioembolic stroke, and for dementia [14,15,16,17,18,25]. A consensus document classified IAB as partial or advanced [16]. A-IAB occurs when the normal interatrial conducction across the Bachmann’s region (Bachmann’s bundle and, ususally, the mid and high septum-LA) is blocked; fibrosis of the Bachmann’s region has been demostrated by using advanced MRI and also in post-mortem specimens [14,26,51,52]. It is expressed in the surface ECG is the presence of P-wave ≥ 120 ms and with a final negative component in the inferior leads (P biphasic morphology in II, III and VF) as a reflection of the retrograde caudo-craneal activation of the LA. Additionally has been described four atypical A-IAB, but all of them have the final negative P-wave component in the lead VF. Both, typical and atypical A-IAB seem to have the same clinical implications and prognostic value.

Nowadays, the focus has been placed on AC as the first step in the thrombogenic cascade that causes cardioembolic stroke [12,51]. A-IAB was present in 26 patients (30.2%): 18 (69.2%) were typical and 8 (30.7%) were atypical A-IAB; Both show a significative relationship with the LA dysfunction, expresed as LA-Sr < 26%, (*p* = 0.032 and *p* = 0.008, respectively) emphasizing the great importance of recognizing these five patterns (Appendix A). IAB is well recognized as a separate entity from LA enlargement [20,22,51]. Although A-IAB is commonly present in the setting of LA dilation. Both are expressed as a P-wave duration ≥ 120 ms and, the LA enlargement, with the presence of a prominent PTFV1 (which has importants limitations because depend on the electrode placement and can be transitoried in dependency to the hemodinamically status) [26,53,54]; the ECG criteria for LA enlargement have good specificity but low sensitivity, In the series, a LA-Sr < 26% was relationated with a dilated LA in the 2D-TTE, but neither with a P-wave > 120 ms nor a incresed PTFV1, making cuestionable a previous AC definition proposed by other authors [32].

Respect to the P-wave voltatge, it depends to the atrial despolarization direcction, the atrial myocardial mass and its conductive properties; when the electrical impulse in the Bachmann’s region is displaced, decreases PVI and P-wave axis (PWA) is displaced; the findings revealed an association between LA dysfunction (LA-Sr < 26%) and a PVI ≤ 0.1 mV (*p* = 0.007) and, although P-wave axis (PWA) was not computed, there was a correlation with A-IAB too (wich always implicates that PWA is abnormal [52]). Both parameters are predictive of the new onset of AF [26,55], howevwer, previous scoring systems for predicting AF did not consider these important P-wave markers. In this way, the recently published MVP ECG risk score (morphology-voltage-P-wave duration) (Appendix A) predicts new-onset of AF but no stroke, and waits for be validated [26,56]. Two considerations can be made regarding the utility of the original MVP score: first, it not include the atypical A-IAB by the P-wave morphology and, second, it requires an semiautomatic digital calipers application to can find differences in P-wave duration of only 20 ms; based on this, patients are classified in three risk groups but, both circunstances implies a potentil misclasification in the real clinical practice [31]. By definition, all five types of A-IAB have the final negative P-wave component in VF lead (and P-waves in leads V5-6 are positives, unlike the junctional rhythm [57]) thus, in the series, MVP score was also calculated basis on a “*modified form*” including all atypical A-IAB. Both scores showed relationship with the LA dysfunction (*p* = 0.038 and *p* = 0.036 respectively), but the “*modified*” MVP score permitted re-clasified 6 patients with intermediate AF risk into the high risk group.

### 4.3. Anticoagulation Therapy versus Aspirin in Patients in Sinus Rhythm

Previous studies examining direct anticoagulant therapy in patients with a history of embolic stroke of undetermined source (ESUS) in sinus rhythm have proven ineffective in reducing the risk of recurrent stroke and systemic embolism [58,59]. In contrast, some studies have demonstrated a decreased risk of stroke and systemic embolism among patients with subclinical atrial fibrillation [60]. Upon a careful analysis of the inclusion criteria, it becomes apparent that there were no specific considerations for TTE and/or ECG markers parameters evaluating LA and/or function, as markers of AC. Consequently, the absence of these criteria raises concerns about the selection of at-risk patients [61].

AC constitutes an intrinsically prothrombotic atrial substrate that may promote AF and thromboembolic events independently of the arrhythmia. The detection of AC should trigger consideration of long-term oral anticoagulation, particularly in patients with a higher thromboembolic risk determined by the CHA_2_DS_2_-VAS_c_ score (until a more rigorously validated scoring system becomes available). 

### 4.4. Limitations of the Study

The study is limited by a small sample size. Currently, we are unable to provide results regarding the potential impact on the risk of stroke. Larger studies are needed to verify this association and explore the potential effectiveness of potential ECG cut-offs and TTE markers for identifying high-risk patients in routine clinical practice, as well as their stratification in the indication of anticoagulants.

## 5. Conclusions

The results showed a significant relationship between LA dysfunction, expressed by an LA-Sr < 26%, and the TTE parameters: (2D-LA-ibV ≥ 34 mL/m^2^; *p* < 0.001), and hypocontractile LA (2D-LA-Ef < 50%; *p* < 0.001). All new AFs were detected in this group. Respect to the ECG, the presence of an A-IAB (*p* = 0.032), a a PVI < 0.1 mV (*p* = 0.007), and a higher MVP ECG score (“*original*”, *p* = 0.038; “*modified*”, *p* = 0.036) were associated with significant LA dysfunction expressed by LA-Sr < 26%.

The independent risk markers of the significative LA dysfunction (LA-Sr < 26%) presence were: 2D-LAEF (AUC: 0.84, CI95%: 0.74–0.94; *p* < 0.001), LA-ibV (AUC: 0.67, CI95%: 0.55–0.80; *p* = 0.008), and A-IAB (AUC: 0.60, CI95%: 0.47–0.73; *p* = 0.109).

Probably, a new simplified ECG score of AC risk, based on the A-IAB and a reduced PVI ≤ 0.1 mV, should be considered to be calculated quickly and easily by the clinicians.

## Figures and Tables

**Figure 1 jcm-12-07315-f001:**
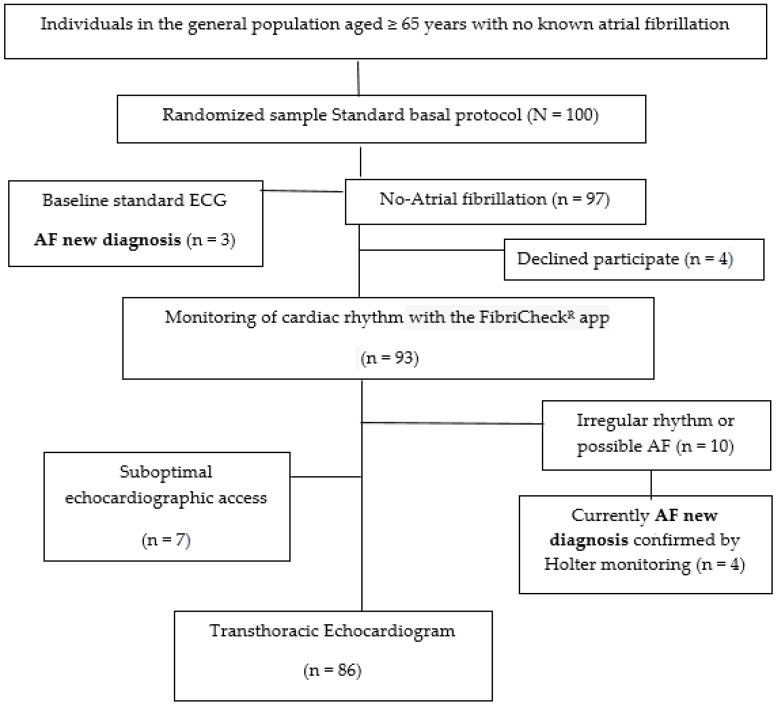
Flowchart Standard basal protocol.

**Figure 2 jcm-12-07315-f002:**
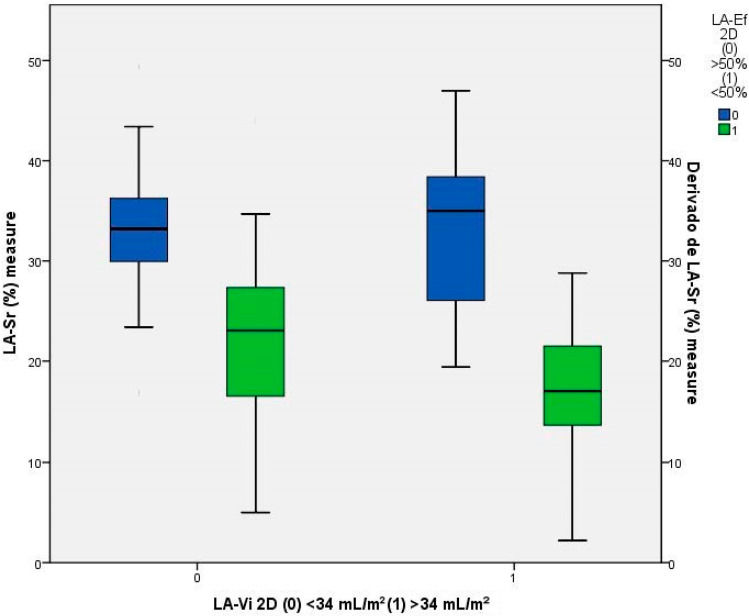
Relationship between LA-Sr and LA-ibV and LAEf in 2D-TTE.

**Figure 3 jcm-12-07315-f003:**
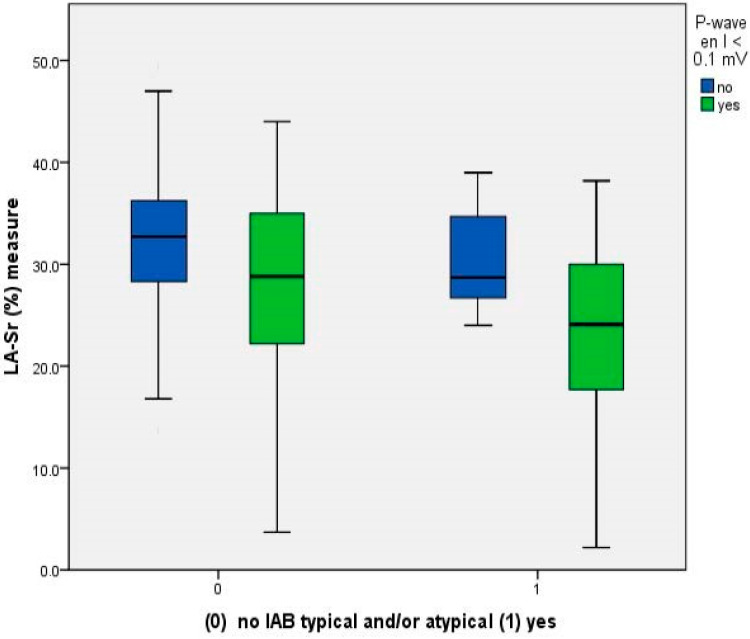
Relationship between LA-Sr, A-IAB and PVI

**Figure 4 jcm-12-07315-f004:**
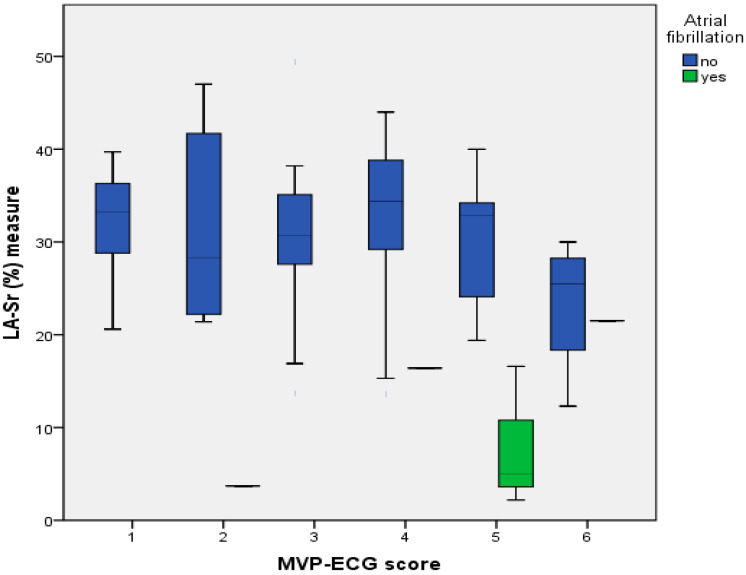
LA-Sr (%) and “*modified*” MVP score according to new AF cases.

**Figure 5 jcm-12-07315-f005:**
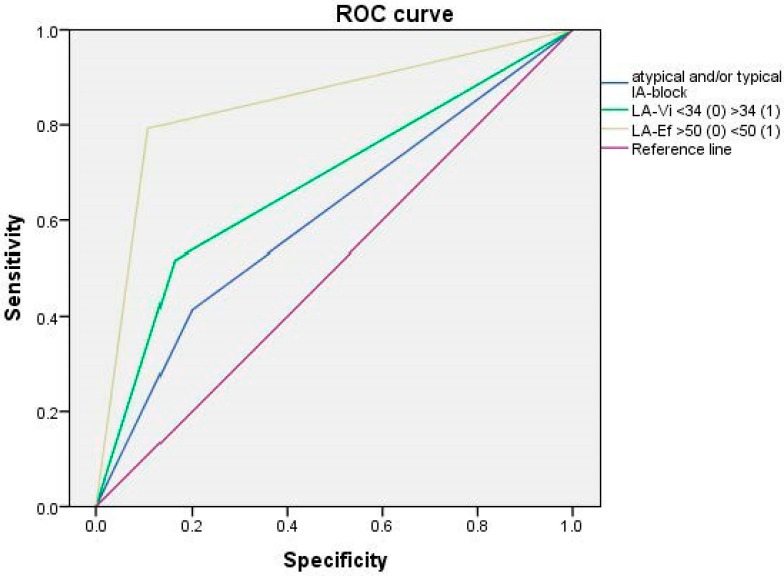
ROC curve.

**Table 1 jcm-12-07315-t001:** Basal characteristics according to LA systolic Strain (LA-Sr) (%).

Variables	All	LA-Sr ≥ 26%	LA-Sr < 26%	*p*
N, %	86 (100.0)	55 (64.0)	31 (36.0)	
Age, $\bar{x}$ ± SD	74.9 ± 5.3	74.1 ± 5.3	76.2 ± 5.6	0.083
Women, n (%)	56 (65.1)	34 (61.8)	22 (70.9)	0.482
Body mass index, $\bar{x}$ ± SD	32.5 ± 5.7	32.5 ± 5.0	32.1 ± 7.6	0.797
Hypertension, n (%)	77 (89.5)	50 (90.9)	27 (87.1)	0.717
Dyslipidemia, n (%)	70 (81.4)	43 (78.2)	27(87.1)	0.394
Diabetes mellitus, n (%)	50 (58.1)	34 (61.8)	16 (51.6)	0.373
Active smoking, n (%)	8 (9.30)	6 (10.9)	2 (6.4)	0.705
Chronic renal failure, n (%)	26 (30.2)	17 (30.9)	9 (29.0)	0.999
Myocardial ischemia, n (%)	10 (11.6)	5 (9.1)	5 (16.1)	0.485
Peripheral vascular disease, n (%)	9 (10.5)	6 (10.9)	3 (9.7)	0.999
Heart failure, n (%)	7 (8.1)	5 (9.1)	2 (6.4)	0.999
CHA_2_DS_2_ VASc score, $\bar{x}$ ± SD	4.1 ± 1.0	4.0 ± 1.0	4.3 ± 1.2	0.251
AF risk [15] ^1^, $\bar{x}$ ± SD	8.6 ± 0.5	8.6 ± 0.5	8.8 ± 0.7	0.087
CVD-SCORE ^2^ ≥ 5%, n (%)	40 (66.6)	24 (66.6)	16 (66.6)	0.658
REGICOR score ^3^, $\bar{x}$ ± SD	6.9 ± 3.9	7.3 ± 3.9	7.1 ± 3.9	0.933
Pfeiffer score, $\bar{x}$ ± SD	0.8 ± 1.1	0.7 ± 0.0	1.0 ± 1.2	0.158
Epworth score, $\bar{x}$ ± SD	5.4 ± 6.1	5.6 ± 6.2	5.2 ± 6.6	0.776
Valvular heart disease, n (%)	4 (4.6)	3 (5.4)	1 (3.2)	0.999
Arterial hypertension, n (%)	74 (86.0)	48 (87.3)	26 (83.9)	0.749
Statins treatment, n (%)	50 (58.1)	35 (63.6)	15 (48.4)	0.181
Diabetes treatment, n (%)	42 (48.8)	29 (52.7)	13 (41.9)	0.375
Antiplatelet drugs, n (%)	24 (27.9)	15 (27.3)	9 (29.0)	0.999
New atrial fibrillation	7	0	7	<0.001

^1^ Reference [36]; ^2^ SCORE2—European High Risk Chart 10-year risk of fatal CVD: high risk and very high risk; ^3^ REGICOR score: Program of the Girona Heart Registry (REGICOR). IMIM, Barcelona. No significant differences were found between the two groups, except for the average LA-Sr (*p* < 0.001), which is the variable used for their classification; regarding the number of new AF diagnoses, all cases were in the group with LA-Sr < 26% (*p* < 0.001).

**Table 2 jcm-12-07315-t002:** Echocardiography results according to LA-Sr (%).

Variables	All	LA-Sr ≥ 26%	LA-Sr < 26%	* p *
N, %	86 (100.0)	55 (64.0)	31 (36.0)	
Age, $\bar{x}$ ± SD	74.9 ± 5.3	74.1 ± 5.3	76.2 ± 5.6	0.083
Women, n (%)	56 (65.1)	34 (61.8)	22 (70.9)	0.482
LV ^1^ Ejection Fraction (%), $\bar{x}$ ± SD	62.6 ± 6.2	63.3 ± 5.6	62.1 ± 7.6	0.411
LV ^1^ Ejection Fraction ≤ 50%, n (%)	2 (2.3)	0 (0.0)	2 (6.4)	na ^10^
LV ^1^ Ejection Fraction ≤ 40%, n (%)	0 (0.0)	0 (0.0)	0 (0.0)	na ^10^
LV ^1^ Diastolic Dysfunction, n (%)	69 (81.2)	45 (81.8)	24 (77.4)	0.622
LV ^1^ Diastolic Dysfunction Grade II-IV, n (%)	11 (12.8)	5 (9.1)	6 (19.3)	0.351
Sigmoid IV ^2^ Septum, n (%)	18 (20.9%)	9 (16.4%)	9 (29.0%)	0.179
Sigmoid IV ^2^ Septum (mm), $\bar{x}$ ± SD	13.1 ± 1.3	12.4 ± 1.0	13.7 ± 1.6	0.026
LA ^3^ vol (mL/m^2^) in 2D-TTE ^4^, $\bar{x}$ ± SD	32.0 ± 9.07	29.4 ± 6.9	36.4 ± 10.4	<0.001
LA ^3^ Dilation, n (%)	24 (28.2)	9 (16.3)	15 (48.4)	0.004
LA ^3^ Dilation: . mild (34–41.9 mL/m^2^) . moderate (42–48 mL/m^2^) . severe (≥48 mL/m^2^)	15 (62.5) 3 (12.5) 6 (25.0)	6 (66.7) 2 (22.2) 1 (11.1)	9 (60.0) 1 (6.7) 5 (33.3)	0.306
LA ^3^ vol (mL/m^2^) in 3D-TTE ^4^, $\bar{x}$ ± SD	38.1 ± 11.0	37.1 ± 11.6	39.9 ± 10.2	0.383
LA ^3^ Ejection Fraction 2D ^8^-TTE ^4^$,\;\bar{x}$ ± SD	50.96 ± 10.08	55.3 ± 5.38	42.4 ± 11.2	<0.001
LA ^3^ Ejection Fraction 3D ^9^-TTE ^4^, $\bar{x}$ ± SD	52.7 ± 13.8	56.2 ± 13.5	44.7 ± 11.4	0.001
LA ^3^ Strain-cd ^5^, $\bar{x}$ ± SD	13.03 ± 5.93	13.1 ± 4.6	11.3 ± 4.4	0.069
LA ^3^ Strain-ct ^6^, $\bar{x}$ ± SD	16.3 ± 7.7	20.4 ± 5.3	8.3 ± 4.5	<0.001
LA ^3^ Strain-r ^7^, $\bar{x}$ ± SD	28.6 ± 9.5	34.2 ± 5.3	18.4 ± 6.2	<0.001
Mitral Regurgitation (grade II–IV), n (%)	19 (22.1)	9 (16.4)	10 (32.2)	0.119
Mitral Stenosis, n (%)	6 (7.0)	2 (3.6)	4 (12.9)	0.179
Aortic Regurgitation (grade II–IV), n (%)	9 (10.5)	5 (9.1)	4 (12.9)	0.823
Aortic Stenosis, n (%)	4 (4.6)	1 (1.8)	3 (9.7)	0.124

^1^ LV: Left Ventricle; ^2^ Sigmoid-shaped Ventricular Septum; ^3^ LA: Left Atrial; ^4^ TTE: Transthoracic Echocardiography; ^5^ Strain-cd: Strain Conduit; ^6^ Strain-ct: Strain Contractile; ^7^ Strain-r: Reservoir Strain; ^8^ 2D-TTE: Two-dimensional Transthoracic Echocardiography; ^9^ 3D-TTE: Three-dimensional Transthoracic Echocardiography. ^10^ na: not applicable.

**Table 3 jcm-12-07315-t003:** Standard ECG Results Based on LA-Sr (%).

Variables	All	LA-Sr ≥ 26%	LA-Sr < 26%	* p *
N, %	86 (100.0)	55 (64)	31 (36)	
Age, $\bar{x}$ ± SD	74.9 ± 5.3	74.1 ± 5.3	76.2 ± 5.6	0.083
Women, n (%)	56 (65.1)	34 (61.8)	22 (70.9)	0.482
Heart rate, average ± SD	69.5 ± 13.0	69.5 ± 13.0	66.0 ± 13.1	0.226
Heart rate < 60 bpm, n (%)	38 (44.2)	25 (45.4)	13 (41.9)	0.823
P-wave > 120 ms, n (%)	64 (74.4)	41 (74.5)	23 (74.2)	0.999
Partial IAB ^1^, n (%)	39 (45.3)	30 (55.5)	9 (29.0)	0.142
Advanced IAB ^2^, n (%)	26 (30.2)	12 (21.8)	14 (45.2)	0.032
Typical advanced IAB, n (%)	18 (20.9)	9 (16.3)	9 (29.0)	0.272
Atypical advanced IAB, n (%)	8 (9.3)	3 (5.4)	5 (16.1)	0.008
P-wave < 0.1 mV lead I, n (%)	43 (50.0)	21 (38.2)	22 (71.0)	0.007
P-wave > 0.04 s in V1, n (%)	58 (67.4)	39 (70.9)	19 (61.3)	0.339
MVP ^3^ score (“*original*”)	3.5 ± 1.4	3.2 ± 1.4	3.9 ± 1.37	0.038
. low: 0–1, n (%) . intermediate, n (%) . high: 5–6, n (%)	19 (22.3) 47 (55.3) 19 (22.3)	14 (25.4) 31(56.4) 10 (18.2)	5 (16.7) 16 (53.3) 9 (30.0)	
MVP ^3^ score_2 (“*modified*”)	3.6 ± 1.5	3.2 ± 1.5	4.0 ± 1.4	0.036
. low: 0–1, n (%) . intermediate, n (%) . high: 5–6, n (%)	19 (22.3) 41 (48.2) 25 (29.4)	14 (25.4) 29 (52.7) 12 (21.8)	5 (16.7) 12 (40.0) 13 (43.3)	
QRS > 120 ms, n (%)	12 (13.9)	8 (14.5)	4 (12.9)	0.999
Sokolow-Lyon and/or Cornell (+) criteria, n (%)	5 (5.8)	2 (3.6)	3 (9.7)	0.349
Atrioventricular block, n (%)	14 (16.3)	9 (16.4)	5 (16.1)	0.999
Premature supraventricular beats, n (%)	5 (5.8)	2 (3.6)	3 (9.7)	0.349

^1^ Partial IAB: Partial interatrial block as *p*-wave duration ≥ 120 milliseconds (ms); ^2^ Advanced IAB: *p*-wave duration ≥ 120 ms with biphasic *p*-wave morphology in the inferior leads; ^3^ MVP score: MVP ECG risk score (morphology–voltage–*p*-wave-duration).

## Data Availability

Data generated or analyzed during this study are not publicly available to preserve individuals’ privacy, but are available from the corresponding author on reasonable request.

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
