# Peer review of "Echocardiography and Electrocardiography in Detecting Atrial Cardiomyopathy: A Promising Path to Predicting Cardioembolic Strokes and Atrial Fibrillation"

_jcm, 2023, doi:10.3390/jcm12237315_

Round 1
Reviewer 1 Report
Comments and Suggestions for Authors
It is unclear if this is a “Study Design” manuscript or a full study results. However they are describing a relatively small cohort.
They claim that they included a randomized sample of 100 patients. Randomized to what? Echo versus not echo?
Figure 1: Some components are in Spanish. For example, No-FA, Estandar basal ECG
Abstract: “Its recognision shoul determinate the patients management in order to prevent cardioembolic strokes” please correct to “recognition” and “should “. This conclusion is not supported by your data and is not the primary aim of the study.
I see no sample size calculations.
If the aim is to show whether using the ECG can predict Atrial Cardiomyopathy by echo, they should present the data this way and show what are the ECG cutoffs that predict LA dilatation and/or LA-SR<26%. Currently, they show that we can use the echo topredict ECG parameters. It makes no sense.
Figure 2 is unclear.
Table 3: I have never seen before a P value above 1
Table 3. What is MVP score? It is not explained in the Methods.
No figure legends.
Discussion: “To the best of our knowledge, this is the first study to describe an association between ECG P-wave abnormalities and 2D-TTE and 3D-TTE LA size and function parameters. Combining LA strain imaging with standard ECG has the potential to open new avenues for early AF diagnosis and the clinical management, and for prevent cardioembolic stroke regardless heart rhythm” Based on what? How many patients had stroke?
The conclusions are not supported by the data presented.
Comments on the Quality of English LanguagePlease translate all to English
Author Response
We are very grateful for the review of our research, as well as for the opportunity to improve the manuscript. Certainly, the issues mentioned are very useful and we believe that they deserve the modifications made on our part. Therefore, please find attached the answers offered according to the points provided, and the new revised manuscript.
The manuscript has been carefully examined, and confirmed for accuracy by a professional translation company with recognized credentials.
- It is unclear if this is a “Study Design” manuscript or a full study results. However they are describing a relatively small cohort.
The expression "study design" refers to the type of study that includes the described data. The choice of study design depends on the research question, objectives, and the nature of the phenomenon being investigated. Perhaps another term could have been used, such as “research design” or “investigative approach”.
The results correspond to the baseline characteristics of the cases included in the ClinicalTrials.gov NCT 05772806 (03/04/2023).
The sample calculation has been included in the methodology.
Currently, there is a lack of evidence regarding the outcomes associated with early diagnosis of AF and potential vascular comorbidities before its formal diagnosis. This entails utilising a combination of high-risk clinical criteria, echocardiography, MRI, and monitoring in patients without a prior AF diagnosis.
Possibly, owing to the innovative nature of exploring ECG variables correlated with new indicators from Transthoracic Echocardiography in patients without atrial fibrillation, the sample size appears to be a modest cohort. The study design had to be adjusted based on the available resources (Health research grants of the Strategic Plan for Research and Innovation in Health (PERIS): Grant PERIS 2022 4R22/031 (SLT021/21/000027). Nevertheless, the sample size is consistent with recently published similar studies, such as:
- Huang, T.; Patrick, S.; Mayer, L.K.; Müller-Edenborn, B.; Eichenlaub, M.; Allgeier, M.; Allgeier, J.; Lehrmann, H.; Ahlgrim, C.; Bohnen, M.; et al. Echocardiographic and Electrocardiographic Determinants of Atrial Cardiomyopathy Identify Patients with Atrial Fibrillation at Risk for Left Atrial Thrombogenesis. Clin. Med.2022, 11, 1332. https://doi.org/10.3390/jcm11051332
- Vera A, Cecconi A, Ximénez-Carrillo Á, Ramos C, Martínez-Vives P, Lopez-Melgar B, Sanz-García A, Ortega G, Aguirre C, Montes Á, Vivancos J, Jiménez-Borreguero LJ, Alfonso F; DECRYTORING study investigators. Left Atrial Strain Predicts Stroke Recurrence and Death in Patients With Cryptogenic Stroke. Am J Cardiol. 2023 Oct 25;210:51-57. doi: 10.1016/j.amjcard.2023.10.001.
- They claim that they included a randomized sample of 100 patients. Randomized to what? Echo versus not echo?
A new sentence has been written:
(lines 110-121): A randomized sample was referred to 100 patients aged ≥ 65 years was recruited from the usual consultations in six primary healthcare (PHC) centres managed by the Catalan Health Institute, and located in Tarragona region (South Catalonia, Spain). All of them were at high risk of developing atrial fibrillation (AF) according to the AF-risk calculator, based on a previously validated predictive model for AF in the general population [15].
[13] Diagnostic Approach of Early Atrial Fibrillation, Silent Stroke and Cognitive Disorder in Patients with High-risk (PREFA-TE). Grant PERIS 2022 4R22/031 (SLT021/21/000027). ClinicalTrials.gov identifier (NCT number): NCT05772806
[14] Palà E, Bustamante A, Clua-Espuny JL, Acosta J, Gonzalez-Loyola F, Santos SD, et al. (2022) Blood-biomarkers and devices for atrial fibrillation screening: Lessons learned from the AFRICAT (Atrial Fibrillation Research In CATalonia) study. PLoS ONE 17(8): e0273571. https://doi.org/ 10.1371/journal.pone.0273571 NCT03188484
[15] Clua-Espuny JL, Muria-Subirats E, Ballesta-Ors J, Lorman-Carbo B, Clua-Queralt J, Palà E, Lechuga-Duran I, Gentille-Lorente D, Bustamante A, Muñoz MÁ, Montaner J; AFRICAT Research Group. Risk of Atrial Fibrillation, Ischemic Stroke and Cognitive Impairment: Study of a Population Cohort ≥65 Years of Age. Vasc Health Risk Manag. 2020 Oct 28;16:445-454. doi: 10.2147/VHRM.S276477.
Lines [160-162]: “A standard surface 12-lead ECG (with a filter setting of 150 Hz, recording speed of 25 mm/s, and calibration of 10 mm/mV) was conducted for all patients”
- Figure 1: Some components are in Spanish. For example, No-FA, Estandar basal ECG
Indeed, the issue has been addressed and resolved.
- Abstract: “Its recognision shoul determinate the patients management in order to prevent cardioembolic strokes” please correct to “recognition” and “should “. This conclusion is not supported by your data and is not the primary aim of the study.
Done. It has been amended
- I see no sample size calculations.
The sample calculation has been included in the methodology.
- If the aim is to show whether using the ECG can predict Atrial Cardiomyopathy by echo, they should present the data this way and show what are the ECG cutoffs that predict LA dilatation and/or LA-SR<26%. Currently, they show that we can use the echo to predict ECG parameters. It makes no sense.
Thank you for your insightful comment and have been highlighted in the Discussion section [lines 378-384]. We acknowledge the importance of aligning our presentation with the primary aim of the study, which is to investigate whether using the ECG can improve the identification of high-risk patients without atrial fibrillation, in whom echocardiography enhances the predictability of future atrial fibrillation and, therefore, the initiation of anticoagulation. Currently, there is a scarcity of longitudinal data linking ECG abnormalities in the P-wave, anomalous echocardiographic results, and the incidence or risk of developing atrial fibrillation (AF). The study is prospective, and its baseline results only allow describing a potential association between these variables. If confirmed, further studies will be necessary. Nevertheless, its potential benefit in the early diagnosis of AF using universally available means such as ECG should be considered.
At the conclusion of the study, with more longitudinal data, we will reassess our data presentation to provide a clear illustration of the ECG cut-offs predicting left atrial dilatation and/or left atrial strain (LA-SR) <26%. Your feedback is invaluable, and we are dedicated to refining our approach to improve the coherence of our findings. Additionally, it would enable more efficient screening of patients with the ultimate goal of assessing its potential impact on stroke prevention.
- Figure 2 is unclear.
The authors agree with the reviewer's opinion, and as it does not provide added visual value, they have decided to remove it.
- Table 3: I have never seen before a P value above 1
Neither the authors. It really appears to be a joke, and we apologize for this mistake, which has been rectified.
- Table 3. What is MVP score? It is not explained in the Methods.
The MVP ECG risk score (morphology-voltage-P-wave duration) was included as a measurement variable in the Methodology (lines 172). Figure 4 has been added concerning risk levels in the MVP test and Left Atrial reservoir Strain (LA-Sr) value, along with a supplementary descriptive Table 1S and reference number 54. Their points are based on P-wave morphology in inferior leads, voltage in lead 1, and P-wave duration (MVP), and it helps to predict new-onset AF.
- No figure legends.
Done
- Discussion: “To the best of our knowledge, this is the first study to describe an association between ECG P-wave abnormalities and 2D-TTE and 3D-TTE LA size and function parameters. Combining LA strain imaging with standard ECG has the potential to open new avenues for early AF diagnosis and the clinical management, and for prevent cardioembolic stroke regardless heart rhythm” Based on what? How many patients had stroke?
Certainly, it is a conclusion not aligned with the current results of the study, which is why it should be reformulated.
- The conclusions are not supported by the data presented.
The conclusions have been reformulated and adjusted to the results described

Reviewer 2 Report
Comments and Suggestions for Authors
The interesting work I hope will shed light on a field that is in the rise- atrial cardiopathy.
In addition to not holding on to each part individually, I think the complete concept of work is excellent. Detailed material and methods displayed. Co-appropriate statistical methods used. Excellent results.
What unnecessarily increases the length of the item is the overwhelming discussion in which there are entire paragraphs belonging to the introduction part. I suggest shortening the part of the discussion concerning the explanation meaning of LA strain, LA systolic and diastolic function. If necessary, that should be moved to introduction.
Same comment about the ECG parameters.
Author Response
Reviewer 2.
The authors are very grateful to the reviewers for their valuable comments, positive feedback, and the opportunity to reconsider the manuscript. They have been diligent in addressing the required comments and suggestions.
The manuscript has been carefully examined, and confirmed for accuracy by a professional translation company with recognized credentials.
- The interesting work I hope will shed light on a field that is in the rise- atrial cardiopathy.
- In addition to not holding on to each part individually, I think the complete concept of work is excellent. Detailed material and methods displayed. Co-appropriate statistical methods used. Excellent results.
Thank you for your positive feedback! It is encouraging to hear that you appreciate the comprehensive approach of the study, including the detailed material and methods, as well as the appropriate statistical methods employed. The excellent results further underscore the strength of the work. It is promising that it might contribute valuable insights to the growing field of atrial cardiopathy
- What unnecessarily increases the length of the item is the overwhelming discussion in which there are entire paragraphs belonging to the introduction part. I suggest shortening the part of the discussion concerning the explanation meaning of LA strain, LA systolic and diastolic function. If necessary, that should be moved to introduction.
- Same comment about the ECG parameters.
The authors have made an effort to comply with suggestions, acknowledging challenges in separating references and results due to the complexity of variables proposed for early atrial fibrillation diagnosis in primary care setting. The Discussion has been substantially reduced and transferred to the Introduction, aiming to integrate its valuable content with the suggestions provided by the reviewer. Nevertheless, we hope to have adequately addressed your suggestions. Changes are in red colour.
